# Symmetry-adapted generation of 3d point sets for the targeted discovery of molecules

**Niklas W. A. Gebauer**
Machine Learning Group
Technische Universität Berlin
10587 Berlin, Germany
`n.wa.gebauer@gmail.com`

**Michael Gastegger**
Machine Learning Group
Technische Universität Berlin
10587 Berlin, Germany
`michael.gastegger@tu-berlin.de`

**Kristof T. Schütt**
Machine Learning Group
Technische Universität Berlin
10587 Berlin, Germany
`kristof.schuett@tu-berlin.de`

## Abstract

Deep learning has proven to yield fast and accurate predictions of quantum-chemical properties to accelerate the discovery of novel molecules and materials. As an exhaustive exploration of the vast chemical space is still infeasible, we require generative models that guide our search towards systems with desired properties. While graph-based models have previously been proposed, they are restricted by a lack of spatial information such that they are unable to recognize spatial isomerism and non-bonded interactions. Here, we introduce a generative neural network for 3d point sets that respects the rotational invariance of the targeted structures. We apply it to the generation of molecules and demonstrate its ability to approximate the distribution of equilibrium structures using spatial metrics as well as established measures from chemoinformatics. As our model is able to capture the complex relationship between 3d geometry and electronic properties, we bias the distribution of the generator towards molecules with a small HOMO-LUMO gap – an important property for the design of organic solar cells.

## 1 Introduction

In recent years, machine learning enabled a significant acceleration of molecular dynamics simulations [1–7] as well as the prediction of chemical properties across chemical compound space [8–13]. Still, the discovery of novel molecules and materials with desired properties remains a major challenge in chemistry and materials science. Accurate geometry-based models require the atom types and positions of candidate molecules in *equilibrium*, i.e. local minima of the potential energy surface. These have to be obtained by geometry optimization, either using computationally expensive quantum chemistry calculations or a neural network trained on both compositional (chemical) and configurational (structural) degrees of freedom [13–16]. On the other hand, bond-based methods that predict chemical properties using only the molecular graph [17–20] do not reach the accuracy of geometry-based methods [11, 21]. In particular, they lack crucial information to distinguish spatial isomers, i.e. molecules that exhibit the same bonding graph, but different 3d structures and chemical properties. As an exhaustive search of chemical space is infeasible, several methods have been proposed to generate molecular graphs [22, 23] and to bias the distribution of the generative model

towards desired chemical properties for a guided search [24, 25]. Naturally, these models suffer from the same drawbacks as bond-based predictions of chemical properties.

In this work, we propose *G-SchNet (Generative SchNet)* for the generation of rotational invariant 3d point sets. Points are placed sequentially according to a learned conditional probability distribution that reflects the symmetries of the previously placed points by design. Therefore, G-SchNet conserves rotational invariance as well as local symmetries in the predicted probabilities. We build upon the SchNet architecture [14] using continuous-filter convolutional layers to model the interactions of already placed points. This allows local structures such as functional groups as well as non-bonded interactions to be captured. Note, that we explicitly do *not* aim to generate molecular graphs, but directly the atomic types and positions, which include the full information necessary to solve the electronic problem in the Born-Oppenheimer approximation. This allows us to avoid abstract concepts such as bonds and rings, that are not rooted in quantum mechanics and rely heavily on heuristics. Here, we use these constructs only for evaluation and comparison to graph-based approaches. In this manner, our approach enables further analysis using atomistic simulations after generation. In particular, this work provides the following main contributions:

- We propose the autoregressive neural network *G-SchNet* for the generation of 3d point sets incorporating the constraints of Euclidean space and rotational invariance of the atom distribution as prior knowledge[1].

- We apply the introduced method to the generation of organic molecules with arbitrary composition based on the QM9 dataset [26–28]. We show that our model yields novel and accurate equilibrium molecules while capturing spatial and structural properties of the training data distribution.

- We demonstrate that the generator can be biased towards complex electronic properties such as the HOMO-LUMO gap – an important property for the design of organic solar cells.

- We introduce datasets containing novel, generated molecules not present in the QM9 dataset. These molecules are verified and optimized at the same level of theory used for QM9[2].

## 2   Related work

Existing generative models for 3d structures are usually concerned with volumetric objects represented as point cloud data [29, 30]. Here, the density of points characterizes shapes of the target structures. This involves a large amount of points where the exact placement of individual points is less important. Furthermore, these architectures are not designed to capture rotational invariances of the modeled objects. However, the accurate relative placement of points is the main requirement for the generation of molecular structures.

Other generative models for molecules resort to bond-based representations instead of spatial structures. They use text-processing architectures on SMILES [31] strings (e.g. [19, 32–41]) or graph deep neural networks on molecular graphs (e.g. [22–25, 42, 43]). Although these models allow to generate novel molecular graphs, ignoring the 3d positions of atoms during generation discards valuable information connected to possible target properties. Mansimov et al. [44] have proposed an approach to sample 3d conformers given a specific molecular graph. This could be combined with aforementioned graph-based generative models in order to generate 3d structures in a two-step procedure. For the targeted discovery of novel structures, both parts would have to be biased appropriately. Again, this can be problematic if the target properties crucially depend on the actual spatial configuration of molecules, as this information is lacking in the intermediate graph representation.

G-SchNet, in contrast, directly generates 3d structures without relying on any graph- or bond-based information. It uses a spatial representation and can be trained end-to-end, which allows biasing towards complex, 3d structure-dependent electronic properties. Previously, we have proposed a similar factorization to generate 3d isomers given a fixed composition of atoms [45]. The G-SchNet architecture improves upon this idea as it can deal with arbitrary compositions of atoms and introduces auxiliary tokens which drastically increase the scalability of the model and the robustness of the approximated distribution.

## 3 Symmetry-adapted factorization of point set distributions

We aim to factorize a distribution over point sets that enables autoregressive generation where new points are sampled step by step. Here, the conditional distributions for each step should be symmetry-adapted to the previously sampled points. This includes invariance to permutation, equivariance to rotation and translation[3] as well as the ability to capture local symmetries, e.g. local rotations of functional groups.

A 3d point set consists of a variable number $n$ of positions $\mathbf{R}_{\leq n}$ and types $\mathbf{Z}_{\leq n}$. Each position $\mathbf{r}_i \in \mathbb{R}^3$ is associated with a type $Z_i \in \mathbb{N}$. When generating molecules, these types correspond to chemical elements. Beyond those, we use auxiliary tokens, i.e. additional positions and types, which are not part of a generated point set but support the generation process. A partially generated point set including $t$ of such auxiliary tokens is denoted as $\mathbf{R}_{\leq i}^t = (\mathbf{r}_1, ..., \mathbf{r}_t, \mathbf{r}_{t+1}, ..., \mathbf{r}_{t+i})$ and $\mathbf{Z}_{\leq i}^t = (Z_1, ..., Z_t, Z_{t+1}, ..., Z_{t+i})$, where the first $t$ indices correspond to tokens and all following positions and types indicate actual points. We define the distribution over a collection of point sets as the joint distribution of positions and types, $p(\mathbf{R}_{\leq n}, \mathbf{Z}_{\leq n})$. Instead of learning the usually intractable joint distribution directly, we factorize it using conditional probabilities

$$p(\mathbf{R}_{\leq n}, \mathbf{Z}_{\leq n}) = \prod_{i=1}^{n} \left[ p\left(\mathbf{r}_{t+i}, Z_{t+i} | \mathbf{R}_{\leq i-1}^t, \mathbf{Z}_{\leq i-1}^t\right) \right] \cdot p(stop | \mathbf{R}_{\leq n}^t, \mathbf{Z}_{\leq n}^t) \tag{1}$$

that permit a sequential generation process analog to autoregressive graph generative networks [22, 24] or PixelCNN [46]. Note that the distribution over the first point in the set, $p(\mathbf{r}_{t+1}, Z_{t+1} | \mathbf{R}_{\leq 0}^t, \mathbf{Z}_{\leq 0}^t)$, is only conditioned on positions and types of the auxiliary tokens and that, depending on the tokens used, its position $r_{t+1}$ might be determined arbitrarily due to translational invariance. In order to allow for differently sized point sets, $p(stop | \mathbf{R}_{\leq n}^t, \mathbf{Z}_{\leq n}^t)$ determines the probability of stopping the generation process.

While the number and kind of tokens can be adapted to the problem at hand, there is one auxiliary token that is universally applicable and vital to the scalability of our approach: the focus point $c_i = (\mathbf{r}_1, Z_1)$. It has its own artificial type and its position can be selected randomly at each generation step from already placed points ($\mathbf{r}_1 \in \{\mathbf{r}_{t+1}, ..., \mathbf{r}_{t+i-1}\}$). We then require that the new point is in the neighborhood of $c_i$ which may be defined by a fixed radial cutoff. This effectively limits the potential space of new positions to a region of fixed size around $c_i$ that does not grow with the number of preceding points. Beyond that, we rewrite the joint distribution of type and position such that the probability of the next position $\mathbf{r}_{t+i}$ depends on its associated type $Z_{t+i}$,

$$p\left(\mathbf{r}_{t+i}, Z_{t+i} | \mathbf{R}_{\leq i-1}^t, \mathbf{Z}_{\leq i-1}^t\right) = p(\mathbf{r}_{t+i} | Z_{t+i}, \mathbf{R}_{\leq i-1}^t, \mathbf{Z}_{\leq i-1}^t) \, p(Z_{t+i} | \mathbf{R}_{\leq i-1}^t, \mathbf{Z}_{\leq i-1}^t) \tag{2}$$

which allows us to sample the type first and the position afterwards at each generation step. Note that, due to the auxiliary focus token, the sampling of types can already consider some positional information, namely that the next point will be in the neighborhood of the focus point.

While the discrete distribution of types is rotationally invariant, the continuous distribution over positions transforms with already placed points. Instead of learning the 3d distribution of absolute positions directly, we construct it from pairwise distances to preceding points (including auxiliary tokens) which guarantees the equivariance properties:

$$p(\mathbf{r}_{t+i} | \mathbf{R}_{\leq i-1}^t, \mathbf{Z}_{\leq i}^t) = \frac{1}{\alpha} \prod_{j=1}^{t+i-1} p(d_{(t+i)j} | \mathbf{R}_{\leq i-1}^t, \mathbf{Z}_{\leq i}^t). \tag{3}$$

Here $\alpha$ is the normalization constant and $d_{(t+i)j} = ||\mathbf{r}_{t+i} - \mathbf{r}_j||_2$ is the distance between the new position and a preceding point or token. As the probability of $\mathbf{r}_{t+i}$ is given in these symmetry-adapted coordinates, the predicted probability is independent of translation and rotation. Transformed to Cartesian coordinates, this leads to rotationally equivariant probabilities of positions in 3d space according to Eq. 3. Since we employ a focus point, the space to be considered at each step remains small regardless of the size of the generated point set.

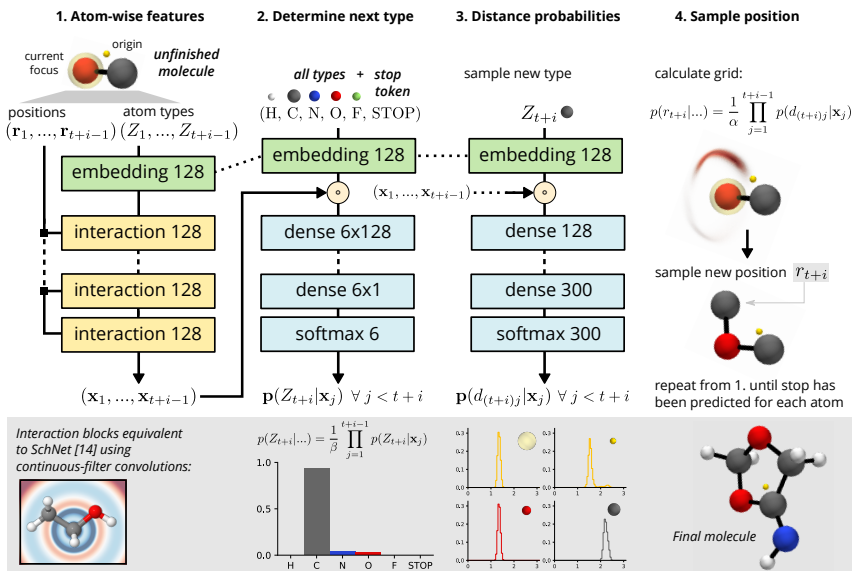

Figure 1: *Scheme of G-SchNet architecture with an exemplary generation step where an atom is sampled given two already placed atoms and two auxiliary tokens: the focus and the origin. The complete molecule obtained by repeating the depicted procedure is shown in the lower right. Network layers are presented as blocks where the shape of the output is given. The embedding layers share weights and the $\odot$-operator represents element-wise multiplication of feature vectors. All molecules shown in figures have been rendered with the 3d visualization package Mayavi [47].*

## 4   G-SchNet

### 4.1   Neural network architecture

In order to apply the proposed generative framework to molecules, we adopt the SchNet architecture originally developed to predict quantum-chemical properties of atomistic systems [5, 14]. It allows to extract atom-wise features that are invariant to rotation and translation as well as the order of atoms. The main building blocks are an embedding layer, mapping atom types to initial feature vectors, and interaction blocks that update the features according to the atomic environments using continuous-filter convolutions. We use the atom representation as a starting point to predict the necessary probability distributions outlined above. Fig. 1 gives an overview of the proposed G-SchNet architecture and illustrates one step of the previously described generation process for a molecule. The first column shows how SchNet extracts atom-wise features from previously placed atoms and the two auxiliary tokens described below in section 4.2. For our experiments, we use nine interaction blocks and 128 atom features.

In order to predict the probability of the next type (Eq. 2), we reuse the embedding layer of the feature extraction to embed the chemical elements as well as the stop token, which is treated as a separate atom type. The feature vectors are copied once for each possible next type and multiplied entry-wise with the corresponding embedding before being processed by five atom-wise dense layers with shifted softplus non-linearity[4] and a softmax. We obtain the final distribution as follows:

$$p(Z_{t+i}|\mathbf{R}^t_{\leq i-1}, \mathbf{Z}^t_{\leq i-1}) = \frac{1}{\beta} \prod_{j=1}^{t+i-1} p\left(Z_{t+i}|\mathbf{x}_j\right). \tag{4}$$

The third column of Fig. 1 shows how the distance predictions are obtained in a similar fashion where the atom features of the sampled type from column 2 are reused. A fully-connected network yields a distribution over distances for every input atom, which is discretized on a 1d grid. For

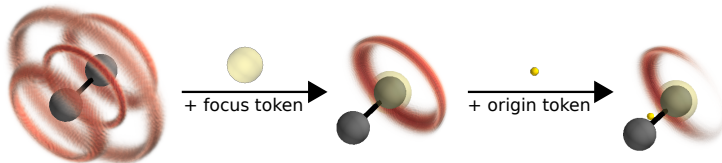

*Figure 2: Effect of the introduced tokens on the distribution over positions of a new carbon atom.*

our experiments, we covered distances up to 15 Å with a bin size of approximately 0.05 Å. The final probability distribution of positions is obtained as described in Eq. 3. We use an additional temperature parameter that allows controlling the randomness during generation. The exact equation and more details on all layers of the architecture are provided in the supplement.

## 4.2 Auxiliary tokens

We make use of two auxiliary tokens with individual, artificial types that are not part of the generated structure. They are fed into the network and influence the autoregressive predictions analog to regular atoms of a molecule. This is illustrated in Fig. 2, where we show the distribution over positions of a new carbon atom given two already placed carbon atoms with and without the placement of additional tokens. Suppose the most probable position is next to either one of the already placed atoms, but *not* between them. Then, without tokens, both carbon atoms predict the same distance distribution with two peaks: at a distance of one bond length and at the distance of two bond lengths to cover both likely positions of the next atom. This leads to high placement probabilities between the atoms as an artifact of the molecular symmetry. This issue can be resolved by the earlier introduced focus point $c_i$. Beyond localizing the prediction for scalability, the focus token breaks the symmetry as the new atom is supposed to be placed in its neighborhood. Thus, the generation process is guided to predict only one of the two options and the grid distribution is free of phantom positions (Fig. 2, middle). Consequently, the focus token is a vital part of G-SchNet.

A second auxiliary token marks the origin around which the molecule grows. The origin token encodes global geometry information, as it is located at the center of mass of a molecule during training. In contrast to the focus token, which is placed on the currently focused atom for each placement step, the position of the origin token remains fixed during generation. The illustration on the right-hand side of Fig. 2 shows how the distribution over positions is further narrowed by the origin token. We have carried out ablation studies where we remove the origin token from our architecture. These experiments are detailed in the supplement and have shown that the origin token significantly improves the approximation of the training data distribution. It increases the amount of validly generated structures as well as their faithfulness to typical characteristics of the training data.

## 4.3 Training

For training, we split each molecule in the training set into a trajectory of single atom placement steps. We sample a random atom placement trajectory in each training epoch. This procedure is described in detail in the supplement. The next type at each step $i$ can then be one-hot encoded by $\mathbf{q}_i^{\text{type}}$. To obtain the distance labels $\mathbf{q}_{ij}^{\text{dist}}$, the distances $d_{(t+i)j}$ between the preceding points (atoms and tokens) and the atom to be placed are expanded using Gaussians located at the respective bins of the 1d distance grid described above. The loss at step $i$ is given by the cross-entropy between labels $\mathbf{Q}_i$ and predictions $\mathbf{P}_i$

$$H\left(\mathbf{Q}_i, \mathbf{P}_i\right) = \underbrace{-\sum_{k=1}^{n_{\text{types}}} \left[\mathbf{q}_i^{\text{type}}\right]_k \cdot \log\left[\mathbf{p}_i^{\text{type}}\right]_k}_{\text{cross-entropy of types}} \underbrace{-\frac{1}{t+i-1} \sum_{j=1}^{t+i-1} \sum_{l=1}^{n_{\text{bins}}} \left[\mathbf{q}_{ij}^{\text{dist}}\right]_l \cdot \log\left[\mathbf{p}_{ij}^{\text{dist}}\right]_l}_{\text{average cross-entropy of distances}},$$

where $\mathbf{p}_i^{\text{type}} = \frac{1}{\beta} \prod_{j=1}^{t+i-1} \mathbf{p}(Z_{t+i}|\mathbf{x}_j)$ with normalizing constant $\beta$ is the predicted type distribution and $\mathbf{p}_{ij}^{\text{dist}} = \mathbf{p}(d_{(t+i)j}|\mathbf{x}_j)$ is the predicted distance distribution. For further details on the training procedure, please refer to the supplement.

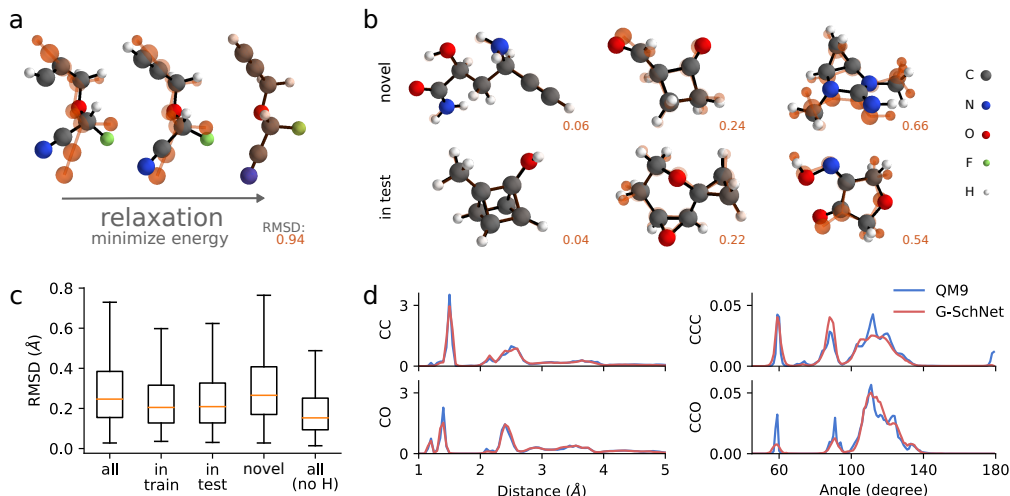

*Figure 3: (a) Scheme showing the relaxation of a molecule along with the resulting root-mean-square deviation (RMSD) between its atomic positions before and after relaxation. The equilibrium structure is indicated by orange shadows. (b) Examples of generated molecules which are novel (top) or resemble test data (bottom). The RMSD increases from left to right. As before, the orange shadows indicate the corresponding equilibrium configurations. (c) Boxplots of the RMSD for generated molecules divided into different sets. The boxes extend from the lower to the upper quartile and the whiskers reach up to 1.5 times the interquartile range. (d) Radial distribution functions for carbon-carbon and carbon-oxygen atom pairs (left) and angular distribution functions for bonded carbon-carbon-carbon and carbon-carbon-oxygen chains (right) in the training data and in generated molecules.*

## 5 Experiments and results

We train G-SchNet on a randomly selected subset of 50k molecules from the QM9 benchmark dataset [26–28] consisting of ~134k organic molecules with up to nine heavy atoms from carbon, nitrogen, oxygen, and fluorine. We use 5k molecules for validation and early stopping while the remaining data is used as a test set. All models are trained with the ADAM optimizer [48]. To implement G-SchNet, we build upon the SchNetPack framework [49] which uses PyTorch [50] and ASE [51].

After training, we generate 20k molecules. In our analysis, we filter generated molecules for *validity*, i.e. we only consider structures that have no disconnected parts and where the number of bonds is equal to the valence of the respective atom type for all atoms. We use Open Babel [52] to obtain (kekulized) bond orders of generated structures for the valency check. In our experiments, about 77% of generated molecules are valid. For the statistics presented in the following, we also filter non-unique structures removing approximately 10%. A step-by-step description of the generation process, details on the subsequent matching of molecules, and tables with extended statistics of G-SchNet and related generative models are given in the supplement.

### 5.1 Accuracy of generated molecules

Unlike molecular graphs which implicitly assume a structural equilibrium, G-SchNet is not restricted to a particular configuration and instead learns to generate equilibrium molecules purely by training on the QM9 data set. In order to assess the quality of our model, we need to compare the generated 3d structures to their relaxed counterparts, i.e. the closest local minimum on the potential energy surface. The latter are found by minimizing the energy with respect to the atom positions at the same level of theory that was used for the QM9 dataset (B3LYP/6-31G(2df,p) [53–56] with the Gaussian version of B3LYP) using the Orca quantum chemistry package [57]. Afterwards, we calculate the root-mean-square deviation (RMSD) of atomic positions between generated and relaxed, ground-truth geometries. The relaxation process is shown in Fig. 3a for an example molecule.

Fig. 3b shows examples of *unseen* generated molecules that are not in the training set with low, average, and high RMSD (left to right). The molecules in the middle column with an RMSD around 0.2 Å closely match the equilibrium configuration. Even the examples with high RMSD do not change excessively. In fact, most of the generated molecules are close to equilibrium as the error distribution shown in Fig. 3c reveals. It shows boxplots for molecules which resemble training data (*in train*), unseen test data (*in test*), and *novel* molecules that cannot be found in QM9. We observe that unseen test structures are generated as accurately as training examples with both exhibiting a median RMSD around 0.21 Å. The test error reported by Mansimov et al. [44] for their molecular graph to 3d structure translation model is almost twice as high (0.39 Å without and 0.37 Å with force field post-processing). However, these results are not directly comparable as they do not relax generated structures but measure the median of the mean RMSD between 100 predicted conformers per graph and the corresponding ground-truth configuration from QM9. For the novel geometries, the RMSDs depicted in the boxplot are slightly higher than in the training or test data cases. This is most likely caused by large molecules with more than nine heavy atoms, which make up almost one-third of the novel structures. Overall the RMSDs are still sufficiently small to conclude that novel molecules are generated close to equilibrium. The last boxplot shows RMSDs considering only heavy atoms of all generated molecules. It is significantly lower than the RMSDs of all atoms, which suggests that the necessary relaxation is to a large part concerned with the rearrangement of hydrogens.

To judge how well our model fits the distribution of QM9, we compare the radial distribution functions of the two most common heavy atom pairs (carbon-carbon and carbon-oxygen) in Fig. 3d. They align well as our training objective directly penalizes deviations in the prediction of distances. Beyond that we plotted angular distribution functions which are not directly related to the optimization objective. They show the distribution over angles of bonded carbon-carbon-carbon and carbon-carbon-oxygen chains. While our model produces smoother peaks here, they also align well indicating that the angles between atoms are accurately reproduced. This is remarkable insofar that only distances have been used to calculate the probabilities of positions.

## 5.2 Structural properties of generated molecules

As proposed by Liu et al. [22], we calculate the atom, bond, and ring counts of the generated molecules to see if G-SchNet captures these structural statistics of the training data. We use the same sets of valid and unique molecules as for the spatial analysis above. The atom and bond counts are determined with Open Babel. For ring counts, the canonical SMILES representation of generated molecules is read with RDKit [58] and the symmetrized smallest set of smallest rings is computed.

We compute the statistics for QM9, molecules generated with G-SchNet as well as molecules obtained by the CGVAE model of Liu et al. [22], a graph-based generative model for molecules (structures provided by the authors). The CGVAE model incorporates valency constraints in its generation process, allowing it to sample only valid molecules. This cannot easily be transferred to 3d models since there is no direct correspondence to bond order for some spatial arrangements. This is because the associated bond order may depend on the not yet sampled atomic environment. However, the valency check is a comparably fast post-processing step that allows us to remove invalid molecules at low cost. Furthermore, in contrast to our approach, CGVAE does not actively place hydrogen atoms and uses all available structures from QM9 for training.

Fig. 4 shows the statistics for the QM9 dataset (first bars of each plot), G-SchNet (second bars), as well as CGVAE (third bars). We observe that G-SchNet captures the atom and bond count of the dataset accurately. CGVAE shows comparable results with a slightly increased heavy atom and bond count. The ring count of CGVAE is significantly higher than in the training data distribution, especially concerning five- and six-membered rings. The average number of rings per molecule is better reproduced by G-SchNet, but here we observe an increase in three- and four-membered rings.

As illustrated in Fig. 2, approximating positions in terms of distances can lead to symmetry artifacts at wrong bond angles if no auxiliary tokens are used. In order to rule out that the increase of three- and four-membered rings is an artifact of our model, we filter all molecules containing such rings from QM9. We train G-SchNet on the resulting subset of approximately 52k structures, where we use 50k molecules for training and the remaining 2k for validation. If our model was biased towards small rings, we expect to find them in the generated molecules regardless. We generate 20k molecules with this model and keep structures that are both valid (77% of all generated) and unique (89% of all

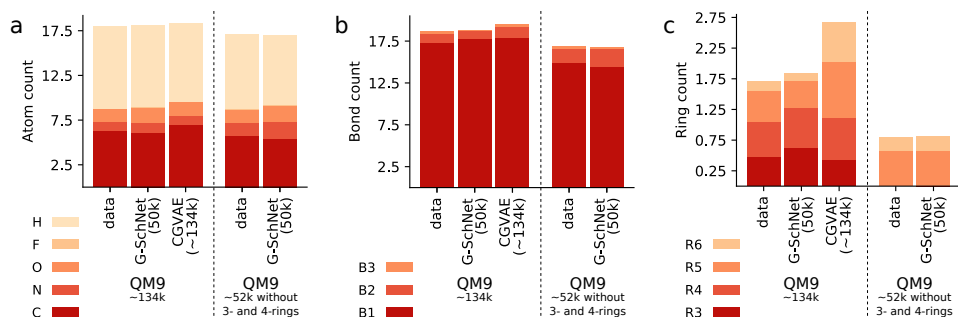

*Figure 4: Bar plots showing the average numbers of atoms, bonds, and rings per molecule. Left-hand side of plots: QM9 dataset, generated molecules of G-SchNet, and generated molecules of CGVAE [22]. Right-hand side of plots: QM9 subset without 3- and 4-rings and generated molecules of G-SchNet trained on that subset. Numbers below model names represent the amount of training data used. B1, B2, and B3 correspond to single, double, and triple bonds. R3, R4, R5, R6 are rings of size 3 to 6.*

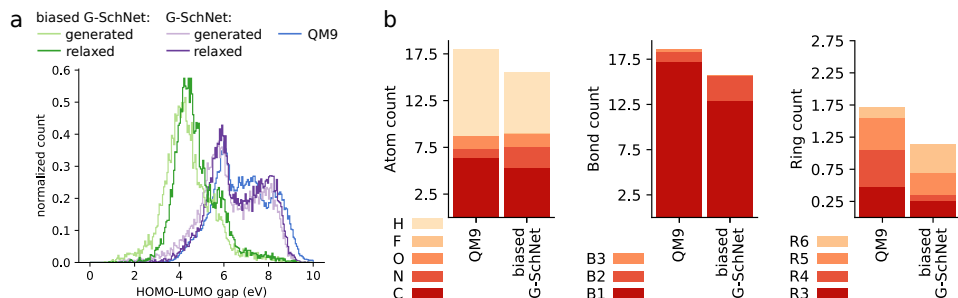

*Figure 5: (a) Histograms of calculated HOMO-LUMO gaps for molecules generated with the biased G-SchNet (green curves), G-SchNet before biasing (purple curves), and for the QM9 dataset. (b) Bar plots showing the average numbers of atoms, bonds, and rings per molecule for QM9 and for molecules generated with the biased G-SchNet. B1, B2, and B3 correspond to single, double, and triple bonds. R3, R4, R5, R6 are rings of size 3 to 6.*

valid). The right-hand side of each bar plot in Fig. 4 shows the corresponding results, comparing the statistics of the QM9 subset (fourth bars) with those of the generated molecules (fifth bars). Again, the atom and bond count statistics are accurately matched. More importantly, the generated molecules almost perfectly resemble the ring count of the training data, exhibiting no three- or four-membered rings. This hints at the possibility to bias the predicted distribution by constraining the training data.

### 5.3 Targeted discovery of molecules with small HOMO-LUMO gaps

Finally, we aim to guide the generator towards a desired value range of a complex electronic property, namely a small HOMO-LUMO gap. This property describes the energy difference between the highest occupied and lowest unoccupied molecular orbital and is an important measure for designing organic semiconductors, an essential component of e.g. organic solar cells and OLED displays. To this end, we filter all ~3.8k equilibrium structures from QM9 with a HOMO-LUMO gap smaller than 4.5 eV. Then we fine-tune the G-SchNet model previously trained on all 50k molecules of the training set on this small subset. We use 3.3k structures for training and the remaining 0.5k for validation and early stopping. The optimizer is reset to its initial learning rate and the optimization protocol for biasing is the same as for regular training.

We generate 20k molecules with this fine-tuned model. The validity decreases to 69% of which 74% are unique. This drop in validity and uniqueness is expected as the amount of generated molecules is significantly higher than the amount of available training structures with the target property. We relax the valid and unique molecules using the same level of theory as described in section 5.1.

Fig. 5a shows histograms of calculated HOMO-LUMO gaps for generated molecules before and after relaxation. There is a significant shift to smaller values compared to the histograms of HOMO-LUMO gaps of the full QM9 dataset. Before biasing, only 7% of all validly generated molecules had a HOMO-LUMO gap smaller than 5.0 eV. The fine-tuning raises this number to 43%. We observe in Fig. 5b how the structural statistics of molecules generated with the biased G-SchNet deviate from the statistics of all molecules in QM9. Compared to the original dataset, the new molecules exhibit an increased number of double bonds in addition to a tendency towards forming six-membered cycles. These features indicate the presence of conjugated systems of alternating single and double bonds, as well as aromatic rings, both of which are important motifs in organic semiconductors.

## 6 Conclusions

We have developed the autoregressive G-SchNet for the symmetry-adapted generation of rotationally invariant 3d point sets. In contrast to graph generative networks and previous work on point cloud generation, our model both incorporates the constraints of euclidean space and the spatial invariances of the targeted geometries. This is achieved by determining the next position using distances to previously placed points, resulting in an equivariant conditional probability distribution. Beyond these invariances, local symmetries are captured by means of point-wise features from the SchNet architecture for atomistic predictions.

We have shown that G-SchNet is able to generate accurate 3d equilibrium structures of organic molecules. The generated molecules closely resemble both spatial and structural distributions of the training data while our model yields 79% unseen molecules. On this basis, we have introduced a dataset of >9k novel organic molecules that are not contained in QM9. Finally, we fine-tuned G-SchNet on a small subset of molecules to bias the distribution of the generator towards molecules with small HOMO-LUMO gap – an important electronic property for the design of organic solar cells. We have generated a dataset of >3.6k novel structures with the desired property, demonstrating a promising strategy for targeted discovery of molecules. Further experiments documented in the supplementary material demonstrate that G-SchNet can be biased towards other electronic properties in the same manner.

Directions for future work include scaling to larger systems, a direct conditioning on chemical properties as well as including periodic boundary conditions. We expect G-SchNet to be widely applicable, for instance, to the guided exploration of chemical space or crystal structure prediction.

**Acknowledgments**

We thank Liu et al. [22] for providing us with generated molecules of their CGVAE model. This work was supported by the Federal Ministry of Education and Research (BMBF) for the Berlin Center for Machine Learning (01IS18037A). MG was provided financial support by the European Unions Horizon 2020 research and innovation program under the Marie Skłodowska-Curie grant agreement NO 792572. Correspondence to NWAG and KTS.

## Footnotes

[1]The code is publicly available at `www.github.com/atomistic-machine-learning/G-SchNet`

[2]The datasets are publicly available at `www.quantum-machine.org`.

[3]If we consider a point set *invariant* to rotation and translation, the conditional distributions have to be *equivariant* to fulfill this property.

[4]as used in SchNet [14]: $\text{ssp}(x) = \ln\left(0.5e^x + 0.5\right)$

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
