[Supplementary Material]

# Supplementary material: Symmetry-adapted generation of 3d point sets for the targeted discovery of molecules

## Architecture

Here we summarize the exact settings used in all layers of our neural network architecture. The structure of the interaction blocks can be found in Schütt et al. [14]. The number of atom features was set to 128 and used in all atom-wise dense layers of the interaction block and filter-generating layers. Distances are expanded in the filter-generating layers using 25 Gaussians with equally spaced centers $0\,\text{Å} \leq \mu \leq 10\,\text{Å}$. Overall, we use nine interaction blocks for feature extraction. We re-use one embedding layer with 128 features at all steps depicted in Fig. 1 of the paper. The output network for type predictions consists of five atom-wise dense layers with shifted-softplus non-linearity and 128, 96, 64, 32, and 1 atom features, respectively. The output network for distance predictions also consists of five atom-wise dense layers with shifted-softplus non-linearity and 128, 171, 214, 257, and 300 atom features. Both output networks contain a final softmax layer.

## Training details

The neural networks were trained with stochastic gradient descent using the ADAM optimizer [48]. We start with a learning rate of $10^{-4}$ which is reduced using a decay factor of $0.5$ after 10 epochs without improvement of the validation loss. The training is stopped at lr $\leq 10^{-6}$. Afterwards, the model with lowest validation error is selected for generation.

While the atom type labels $\mathbf{q}_i^{\text{type}}$ can be directly obtained from the training data, the labels for distance distributions $\mathbf{q}_{ij}^{\text{dist}}$ are obtained using:

$$\left[\mathbf{q}_{ij}^{\text{dist}}\right]_l = \frac{\exp(-\frac{1}{\gamma}(d_{(t+i)j} - \mu_l)^2)}{\sum_{l=1}^{300}\exp(-\frac{1}{\gamma}(d_{(t+i)j} - \mu_l)^2)} \qquad \forall\, j < t + i.$$

The width of the Gaussians can be tuned with the $\gamma$ parameter, which we set to 10% of the bin size in our experiments, resulting in very peaky, uni-modal label vectors.

## Controlling randomness with the temperature parameter

In order to control the randomness during generation, we do not directly implement Eq. 3 but include a temperature parameter $T$:

$$p(\mathbf{r}_{t+i}|\mathbf{R}_{\leq i-1}^t, \mathbf{Z}_{\leq i}^t) = \frac{1}{\alpha}\exp\left(\frac{\sum_{j=1}^{t+i-1}\log p(d_{(t+i)j}|\mathbf{x}_j)}{T}\right). \qquad (5)$$

Increasing $T$ will smoothen the grid distribution, effectively increasing randomness, whereas small values lead to a peaky distribution and thus less randomness. For all experiments, we chose a fixed temperature of $T = 0.1$ according to the following procedure.

We used the G-SchNet model trained on 50k equilibrium structures from QM9 [26–28] and generated 20k molecules for each $T \in \{2, 1, 0.1, 0.01, 0.001\}$. From each set, 1k valid and unique molecules were randomly chosen, where 800 resembled test structures, 100 resembled training structures, and the remaining 100 were novel structures with more than 9 heavy atoms. We relaxed the five sets of 1k molecules at the PBE/def2-SVP level of theory[59, 60] using the Orca program, employing the resolution of identity (RI) approximation[61, 62] and Grimme D3 dispersion correction with Becke-Johnson damping.[63] The root-mean-square deviation (RMSD) between atomic positions before and after relaxation was measured. A smaller RMSD means that the generated structures are closer to a true equilibrium configuration.

In Fig. 6 we show boxplots of the RMSD for the five different temperatures $T$. We see that the smallest median values and interquartile ranges are observed for values of $T$ smaller than 1. Since increasing $T$ corresponds to increasing randomness during sampling, the increase in the RMSD for $T = 1$ and $T = 2$ is expected. However, decreasing $T$ to smaller values than $0.1$ does not lead to smaller RMSDs. Generally, we expect the number of unique sampless to decrease as $T$ gets too small. Therefore, for the other experiments, we chose the highest value for $T$ that still produces structures close to equilibrium, namely $T = 0.1$.

*Figure 6: The boxplots show the RMSD of atomic positions between generated and relaxed structures for different values of $T$, where the boxes extend from the lower to the upper quartile values, the red lines indicate the medians, and the whiskers reach up to 1.5 times the interquartile ranges. Outliers are not shown.*

**Sampling generation traces for training**

The generation traces start with the focus and origin tokens set to the same position, which is the center of mass of the respective training molecule. The first new type and position are taken from the atom closest to the center of mass. At each subsequent step, we randomly select one of the already placed non-token atoms as focus point (which is only a single choice for the second step). The new position and type are then taken from the neighbor which is closest to the origin token, where neighbors are all unplaced atoms of the training molecule that are connected to the focus by a bond. If no unplaced neighbors are left, the new type is set to the stop token and the focused atom is marked as finished, i.e. it cannot be chosen as focus anymore. For the next step, another already placed (unfinished) atom is randomly chosen as focus and the procedure is repeated until all atoms have been placed and marked as finished.

**Generating molecules**

For the first step of molecule generation, the focus and origin token are set to the origin of a 3d grid. The grid extends up to 1.7 Å into all dimensions with a step-size of 0.05 Å. The token are processed by G-SchNet to sample the type of the first non-token atom and to obtain the two predicted distance distributions. The probabilities of the grid cell positions are calculated according to Eq. 5 in order to sample the position of the first atom. Then, at each subsequent step, a random unfinished non-token atom is selected as focus token (which is only a single choice for the second step). The grid is centered on the focus (but the origin token stays at the former origin of the grid) and G-SchNet predictions are obtained to sample the type and position of the next atom (see Fig. 1 in the paper for an exemplary generation step). If the stop token is predicted as next type, the currently focused atom is marked as finished and no position is sampled. Instead the next generation step is initialized by randomly selecting one of the remaining unfinished atoms as focus point. The generation process stops if no unfinished atoms are left. For our experiments, we also stopped the process if molecules were not finished after placing 35 atoms and discarded these structures as invalid (usually ~1% of generated molecules).

**Matching molecules**

To remove non-unique structures and identify moleclues that resemble training or test data, we calculate the Tanimoto similarity of path-based fingerprints (FP2) [52] for pairs of molecules with Open Babel. If the similarity is one, we compare the canonical SMILES representations in a second step. If they match, the two molecules are treated as equal. Note that this is a conservative approach as it filters out some spatial isomers which cannot be distinguished with SMILES.

| | | | | | | with origin token | without origin token |
|---|---|---|---|---|---|---|---|
| valid | unique | in train | in test | novel | ≤9 heavy | | |
| ✓ | | | | | | 77.0% | 57.9% |
| ✓ | ✓ | | | | | 70.6% | 56.7% |
| ✓ | ✓ | ✓ | | | | 9.4% | 1.9% |
| ✓ | ✓ | | ✓ | | | 12.0% | 2.6% |
| ✓ | ✓ | | | ✓ | | 49.2% | 52.2% |
| ✓ | | | | | ✓ | 60.4% | 13.7% |

Figure 7: *Ablation study on the effect of the origin token. Statistics are compared for 20k molecules generated by a G-SchNet model with origin token and 20k molecules generated by a G-SchNet model without origin token. Table (a) shows the percentage of generated molecules for which the properties indicated by the check marks in each row hold. The average numbers of atoms, bonds, and rings per generated molecule and per QM9 molecule are compared in (b), (c), and (d), respectively. B1, B2, and B3 correspond to single, double, and triple bonds. R3, R4, R5, and R6 are rings of size 3 to 6.*

### Ablation study

In order to assess the effect of the origin token, we conduct an ablation study where we remove the origin token from the input to a G-SchNet model during training and generation. All other hyperparameters are identical to the ones used when training the standard G-SchNet with origin token. After training, we generate 20k molecules with each architecture and compare their statistics. Fig. 7a shows that the validity of generated molecules drops by almost 20 percent without the origin token. Furthermore, the amount of generated molecules that match QM9 training or test structures significantly decreases. All QM9 structures consist of at most 9 heavy atoms but only 13.7% of the molecules generated without origin token have 9 or less heavy atoms (compared to 60.4% with origin token). The diverging atom count can also be seen in Fig. 7b. Moreover, the bond count in Fig. 7c also diverges from the training data distribution. The ring count in Fig. 7d, on the other hand, is not noticeably better or worse without origin token. There is an increase in six-membered rings and a decrease in all smaller ring structures compared to the model with origin token. Both models slightly diverge from the QM9 training data ring count. Overall, we conclude that the origin token has a significant, positive effect on the approximated probability distribution. It enables G-SchNet to better capture the characteristics of the training data, leading to a model that generates more valid molecules which are more faithful to the training structures but still equally unique and unseen (in test or novel) compared to a model without origin token.

### Targeted discovery with respect to further electronic properties

In a similar fashion to the experiments for molecules with a small HOMO-LUMO gap, we bias G-SchNet models towards large values of three more electronic properties available in QM9, namely the isotropic polarizability, the dipole moment, and the electronic spatial extent. The results depicted in Fig. 8 show clear shifts in the distribution of targeted properties for molecules generated with biased G-SchNets. We again use small subsets of molecules from QM9 exhibiting the respective property for fine-tuning of the G-SchNet model previously trained on 50k structures. These subsets consist of $2100/500$ molecules with an isotropic polarizability $\geq 91$ Bohr$^3$, $3000/500$ molecules with a dipole moment $\geq 5.75$ Debye, and $4400/500$ molecules with an electronic spatial extent $\geq 1785$ Bohr$^2$ for training/validation, respectively. In contrast to the HOMO-LUMO gap experiments, where we relaxed generated structures and calculated the gap numerically with time-consuming DFT simulations, we train three separate SchNet models to predict the three electronic properties. We use 100k molecules from QM9 for training, 10k for validation, and the remaining structures as a test set. The mean absolute test error is $0.070$ Bohr$^3$ for the electronic polarizability, $0.016$ Debye for the dipole moment, and $0.126$ Bohr$^2$ for the electronic spatial extent.

### Detailed statistics

We provide two tables which report more detailed statistics on relevant properties of generated molecules. In Table 1 we compare 20k molecules generated by our standard G-SchNet model, by the G-SchNet model trained on QM9 without structures with 3- or 4-membered rings, by the

*Figure 8: Distribution of three quantum-chemical properties for molecules from QM9 (blue), generated by an unbiased G-SchNet (purple), and generated by G-SchNets biased towards larger values of the respective property (green).*

*Table 1: Statistics for all of our models and CGVAE [22] (molecules provided by the authors). CGVAE was trained on all molecules in QM9. Our models were trained on 50k randomly selected molecules from QM9. For the second model molecules with three- and four-membered rings were excluded. The third model was fine-tuned on 3k molecules with a HOMO-LUMO gap $\leq$ 4.5 eV. The numbers are percent of all validly generated molecules (which are 77%, 78%, 69% and 100% of 20k generated for the models from left to right). The molecules are categorized according to the checkmarks in the first five columns. Unseen refers to molecules not found in the training data and novel stands for molecules not found in QM9. $\leq$9 heavy marks molecules with nine or less heavy atoms and gap refers to the HOMO-LUMO gap and was calculated for relaxed structures.*

| unique | unseen | novel | $\leq$9 heavy | gap $\leq$5eV | G-SchNet 50k | G-SchNet 50k no R3/R4 | G-SchNet 50k + 3k gap$\leq$4.5eV | CGVAE ~134k |
|---|---|---|---|---|---|---|---|---|
| ✓ | | | | | 91.6% | 89.4% | 73.8% | 98.4% |
| ✓ | ✓ | | | | 79.4% | 68.3% | 66.8% | 87.4% |
| ✓ | | ✓ | | | 63.8% | 68.3% | 57.5% | 87.4% |
| ✓ | | | ✓ | | 70.2% | 58.2% | 57.8% | 30.0% |
| ✓ | ✓ | | ✓ | | 58.0% | 37.1% | 50.8% | 18.9% |
| ✓ | | ✓ | ✓ | | 42.4% | 37.1% | 41.5% | 18.9% |
| ✓ | ✓ | | | ✓ | 6.8% | 21.2% | 43.4% | – |

G-SchNet model biased towards small HOMO-LUMO gaps, and by the constrained graph variational autoencoder (CGVAE). The provided numbers are percent of all validly generated structures.

In Table 2 we compare the number of valid, novel, and unique molecules generated by the standard G-SchNet as well as six related generative models that rely either on graphs or SMILES strings as molecule representation. The shown numbers are the respective percentage of 20k generated molecules for each model. Note that this means that a high novelty score can also come from a low number of valid molecules, as each invalid structure is not included in the training data and therefore novel. Similarly, a high uniqueness score is only interesting if the majority of generated molecules is valid as otherwise the model generates many unique but invalid structures. In general, the methods cannot directly be compared since generating 3d molecular structures is a different task than generating graphs or strings. For example, all of the valid G-SchNet molecules correspond to one proper 3d structure whereas valid graphs and SMILES strings may have no, one, or many different corresponding 3d conformers which cannot easily be found without expensive quantum-chemical simulations.

*Table 2: Percent of valid, novel (not in training data), and unique molecules among 20k structures generated after training on QM9 for G-SchNet and related models working with SMILES or graph representations. Molecules are considered valid if the valency-constraints of all its atoms are met. In order to identify duplicate and novel molecules, we use molecular fingerprints and canonical SMILES strings as explained in section "matching molecules" above. Statistics for graph- and SMILES-based models are taken from Fig. 3 in Liu et al. [22].*

|  | **G-SchNet** 3d structure | **CGVAE**\*[22] graph | **GraphVAE**\*[64] graph | **NeVAE**\*[65] graph | **LSTM**\*[22] SMILES | **CVAE**\*[19] SMILES | **GVAE**\*[32] SMILES |
|---|---|---|---|---|---|---|---|
| **valid** | 77.07% | 100.00% | 61.00% | 98.00% | 94.78% | 10.00% | 30.00% |
| **novel** | 87.47% | 94.30% | 85.00% | 100.00% | 82.98% | 90.00% | 95.44% |
| **unique** | 91.91% | 98.57% | 40.90% | 99.86% | 96.94% | 64.50% | 9.30% |

\*statistics are taken from Fig. 3 in Liu et al. [22]