[Reviews · NeurIPS 2019]

Reviewer 1



This paper introduces an autoregression model based on the SchNet. The technical parts seem correct, but the experiment section lacks baselines, like the state-of-the-art methods [1]. This paper is well written and organized clearly, yet there are still some points need to be clarified by the authors. In Equation (1), c_i is included in R_{\le i}, Z_{\le i}, so I’m wondering if it is redundant. I guess the authors want to highlight the role of c_i as the focus point, but we actually don’t need it in the final equation, as already illustrated in Equation (4) and Figure 1. According to Section 3, it seems like all the nodes are created sequentially, then what about the loop/ring in the graph? How to generate such a structure when we assume the new points are bonded to the focus point only? Or if the generation process will generate the same points twice when moving along the trajectory? In line 195, it implies that generated molecules -> SMILES -> bond counts, then what’s the difference with counting bounds directly on the generated graph? I think the work is a little incremental since the key building blocks come from the SchNet. In addition, the experiment section lacks the baselines like [1]. For the targeted discovery, it would be better to explore more tasks. For example, the remaining 11 tasks in QM9 all share the same molecules, so they can be trained quickly following the same pre-trained model. According to section 5.3, the targeted discovery is obtained by using some heuristics (data filtering), so it would be more convincing to compare the G-SchNet with other baseline methods to show its advantage in this setting. [1] You, J., Liu, B., Ying, Z., Pande, V., & Leskovec, J. (2018). Graph convolutional policy network for goal-directed molecular graph generation. In Advances in Neural Information Processing Systems (pp. 6410-6421).

Reviewer 2



The authors extend the existing SchNet architecture to a generative framework, called G-SchNet, which can generate rotational invariant 3d point sets. The auxiliary tokens introduced by the authors are very interesting, which can give some constraints on the generated points. The paper is a complete work with enough experiments to demonstrate the effectiveness of the proposed method. In all, the paper would have some practical impact on research.

Reviewer 3



[Originality] This paper proposes a new task of generating 3D geometry of molecules. I think the task is original and important for computational chemistry. The underlying generative model is a variant of SchNet that expands the molecule one atom at a time along with its distance to previous atoms. In that regard, the model is similar to GraphRNN (You et al., 2018), but operating over point clouds instead of graphs. The related work is mostly complete, but I think the author should discuss how is their method different from Mansimov et al., 2019, which is also a generative model for 3D molecular geometry. AFAIK, Mansimov et al.'s model only generates 3D geometry, while GSchNet learns to generate both the molecule (atoms and bonds) as well as their 3D geometry (distances). [Quality] The submission is technically sound. The experiment covers both conditional and unconditional generation tasks. Overall the experimental results support their claims. It compares against CGVAE in terms of how well the learned distribution matches the training set (number of different atoms, bonds, etc.). It would be better to also compare the distribution of chemical properties (logP, QED, etc.). [Clarity] Most of the parts are clear. What is unclear to me is 1) what is the relaxation procedure and why it is necessary for evaluation. 2) The advantage and disadvantage of the proposed method compared to Mansimov et al., 2019. [Significance] The paper developed a novel method addressing a important task in chemistry. I think other researchers will be interested in using the method for future research. ===================================================================================================================== Here is my response to the author rebuttal: 1) It is good that the authors provided comparison with Mansimov et al., 2019 (a very important baseline). The result shows that G-SchNet achieves better performance. It would be helpful if the authors can clarify more on the setup of Mansimov et al.'s model. 2) The authors explained about relaxation procedure. I think it would be better to provide some examples to illustrate how this would impact the performance (RMSD) in the future (e.g., in the appendix) 3) I agree with the authors regarding the novelty of the task (generating 3D molecular geometries). Based on these points, I will keep my original scores.

[Author Response · NeurIPS 2019]

We thank the reviewers for their insightful comments. Generally, we noticed that we might not have explained clear enough that generating 3d molecular geometries is an independent, quite novel task and not to be confused with generating molecular graph connectivity. We added the following clarification to the ms: *"Note, that we explicitly do not aim to generate molecular graphs, but the atomic types and positions, which include the full information necessary to solve the electronic problem in the Born-Oppenheimer approximation. Thus, we avoid abstract concepts such as bonds and rings, that are not rooted in quantum mechanics and rely heavily on heuristics. We use these constructs only for evaluation and comparison to graph-based approaches."*

**Rev. #2** ***"Auxiliary tokens"*** We agree that our notation of the focus token was confusing. While explicitly mentioned in Sec. 3, we implicitly treat it as an atom with a special type in Sec. 4. We unified the notation to avoid confusion and clarified the significance of the focus token, as it constrains the space to place the next atom to a small region. The method would be impractical without it, as the space spanned by the grid would have to grow with the molecule. Note that we do only place atoms, not bonds (i.e. edges), as these are artificial constructs based on heuristics, e.g. overlapping covalent radii. Thus, we use bonds and rings only to visualize and compare to graph-based methods. ***"Incremental"*** To our knowledge, we present the *first generative neural network for molecular 3d structures with arbitrary composition*. While SchNet is used as the encoder for partial molecules, the overall G-SchNet architecture is novel. We added further baselines comparing to generation of graphs (Tab. 1) and geometries (Mansimov et al., see Rev #4). ***"Data filtering"*** We bias the distribution sampled from our network by fine-tuning on a subset of molecules with the desired property. This is *transfer learning*, not a heuristic. ***Improvements*** We added baseline methods (Tab. 1) and additional tasks from QM9 for targeted discovery (Fig.1 a-c)

**Rev. #3** ***Improvements*** The focus token is essential to our architecture, as explained above and clarified in the ms. We have added an ablation study for the origin token (Fig. 1 d-g). Without the origin token, the validity of generated molecules significantly drops (almost by 20%) and the amount of generated structures faithful to the training data also decreases substantially.

**Rev. #4** ***"Relaxation"*** Added to ms: *"Unlike molecular graphs which implicitly assume a structural equilibrium, G-SchNet is not restricted to a particular configuration and instead learns to generate equilibrium molecules purely by training on the QM9 data set. In order to assess the quality of our model, we need to compare the generated 3d structures to their relaxed counterparts, i.e. the closest local minimum on the potential energy surface. Those are found by minimizing the energy w.r.t. the atom positions."* ***"Related work"*** The reviewer is correct that Mansimov et al. "translate" from graphs to 3d geometry while G-SchNet generates 3d structures from scratch. Their approach could be coupled with graph generative models but these would still be inherently limited to targeting properties that do not crucially depend on the spatial configuration of molecules. We expanded the related work accordingly. We also added Mansimov et al. as a further baseline, noting that the comparison is to be taken with a grain of salt due to the different approach: While Mansimov et al. achieve median RMSD of 0.39Å (0.37Å with force field post-processing) when translating graphs from QM9, the generated structures from G-SchNet contained in the QM9 test set reach a median RMSD of 0.21Å. ***Improvements*** We clarified the relaxation procedure (see above) and added an ablation study for the origin token (Fig. 1 d-g).

*Table 1: Baseline comparisons added to ms (compressed to fit rebuttal).*

|        | **G-SchNet** 3d structure | **CGVAE**\*[21] graph | **GraphVAE**\*[59] graph | **NeVAE**\*[60] graph | **LSTM**\*[21] SMILES | **CVAE**\*[18] SMILES | **GVAE**\*[31] SMILES |
|--------|---------------------------|-----------------------|--------------------------|-----------------------|-----------------------|-----------------------|-----------------------|
| **valid**  | 77.15% | 100.00% | 61.00% | 98.00%  | 94.78% | 10.00% | 30.00% |
| **novel**  | 87.47% | 94.30%  | 85.00% | 100.00% | 82.98% | 90.00% | 95.44% |
| **unique** | 91.87% | 98.57%  | 40.90% | 99.86%  | 96.94% | 64.50% | 9.30%  |

\*statistics are taken from Fig. 3 in [21]

*Figure 1: Additional experiments added to ms: more targets on QM9 (**a-c**) and ablation study (**d-g**) (compressed to fit rebuttal). (**a-c**) Distribution of three quantum-chemical properties for molecules from QM9 (blue), generated by an unbiased G-SchNet (purple), and generated by G-SchNets biased towards larger values of the respective property (green). (**d-g**) Ablation study on the effect of the origin token.*

[Meta-Review · NeurIPS 2019]

This paper introduces a generative model for 3D representations of small molecules, which takes care to encode known symmetries in the energy functions of such molecules. The architecture is novel if a bit on the incremental side, but the application and baseline evaluations are sensible.